# Comparative proteomic analysis of the hemolymph and salivary glands of *Rhodnius prolixus* and *R. colombiensis* reveals candidates associated with differential lytic activity against *Trypanosoma cruzi* Dm28c and *T. cruzi* Y

Hamilton J. Barbosa[1]*, Yazmin Suárez Quevedo[1], Arlid Meneses Torres[1], Gustavo A. Gaitán Veloza[1], Julio C. Carranza Martínez[1], Daniel A. Urrea-Montes[1], Carlos Robello-Porto[2], Gustavo A. Vallejo[1]*

**1** Laboratorio de Investigaciones en Parasitología Tropical (LIPT), Facultad de Ciencias, Universidad del Tolima, Ibagué, Colombia, **2** Departamento de Bioquímica, Facultad de Medicina, Universidad de la República, Montevideo, Uruguay

* hjbarbosav@ut.edu.co (HJB); gvallejo@ut.edu.co (GAV)

## Abstract

### Background

Immune response of triatomines plays an important role in the success or failure of transmission of *T. cruzi*. Studies on parasite–vector interaction have shown the presence of trypanolytic factors and have been observed to be differentially expressed among triatomines, which affects the transmission of some *T. cruzi* strains or DTUs (Discrete Typing Units).

### Methodology/Principal findings

Trypanolytic factors were detected in the hemolymph and saliva of *R. prolixus* against epimastigotes and trypomastigotes of the Y strain (*T. cruzi* II). To identify the components of the immune response that could be involved in this lytic activity, a comparative proteomic analysis was carried out, detecting 120 proteins in the hemolymph of *R. prolixus* and 107 in *R. colombiensis*. In salivary glands, 1103 proteins were detected in *R. prolixus* and 853 in *R. colombiensis*. A higher relative abundance of lysozyme, prolixin, nitrophorins, and serpin as immune response proteins was detected in the hemolymph of *R. prolixus*. Among the *R. prolixus* salivary proteins, a higher relative abundance of nitrophorins, lipocalins, and triabins was detected. The higher relative abundance of these immune factors in *R. prolixus* supports their participation in the lytic activity on Y strain (*T. cruzi* II), but not on Dm28c (*T. cruzi* I), which is resistant to lysis by hemolymph and salivary proteins of *R. prolixus* due to mechanisms of evading oxidative stress caused by immune factors.

**Data Availability Statement:** The mass spectrometry proteomics data have been deposited to the ProteomeXchange Consortium via the PRIDE partner repository with the dataset identifier PXD048605. Publication DOI: 10.1371/JOURNAL.PNTD.0011452 Project Webpage: http://www.ebi.ac.uk/pride/archive/projects/PXD048605 All other relevant data, including three tables with the raw data of biological activity and protein detection are in supporting information.

**Funding:** This work was supported by Ministerio de Ciencia, Tecnología e Innovación (Minciencias) and the Department of Tolima for funding the project through national convocation 755 of 2016 to HJB. The funders had no role in study design, data collection and analysis, decision to publish, or preparation of the manuscript.

**Competing interests:** The authors have declared that no competing interests exist.

## Conclusions/Significance

The lysis resistance observed in the Dm28c strain would be occurring at the DTU I level. *T. cruzi* I is the DTU with the greatest geographic distribution, from the south of the United States to central Chile and Argentina, a distribution that could be related to resistance to oxidative stress from vectors. Likewise, we can say that lysis against strain Y could occur at the level of DTU II and could be a determinant of the vector inability of these species to transmit *T. cruzi* II. Future proteomic and transcriptomic studies on vectors and the interactions of the intestinal microbiota with parasites will help to confirm the determinants of successful or failed vector transmission of *T. cruzi* DTUs in different parts of the Western Hemisphere.

## Author summary

Some factors can facilitate or prevent *T. cruzi* transmission, i.e. vector immunity. Our work has managed to detect a stronger immune response against *T. cruzi* II in *R. prolixus* saliva and haemolymph, compared to that of *R. colombiensis*. Proteins from both species' saliva and haemolymph were analysed for studying factors which might have been involved in such response; most proteins were detected in both species' haemolymph, thereby indicating common immune mechanisms. Three proteins having oxidative immune activity were only expressed in *R. prolixus*. Lipocalin diversity and abundance predominated in *R. prolixus* saliva; these proteins are involved in nitric oxide metabolism and their role in immunity could be key in host defence against *T. cruzi*. Recognising the components modulating parasite transmission in a vector helps in understanding how such factors act independently and how they would act synergistically against *T. cruzi*, thereby enabling us to establish tools regarding Chaga's disease epidemiology, aimed at predicting *T. cruzi* distribution and creating transmission control mechanisms.

## Introduction

In terms of parasite–vector interactions, four determinants of the transmission of *Trypanosoma cruzi* have been recognized: i) the strain and discrete typing units (DTU) of the parasite, ii) the triatomine species, iii) the cellular and humoral immune response of the vector, and iv) the intestinal microbiota of the insect [1,2,3]. Three of these determinants are related to the vector, which has directed special interest to the study of the tissues and mechanisms associated with the insect's immune response, involving the hemolymph, hemocytes, fat bodies, digestive tract, and salivary glands [2,4]. Vectors have an innate immune system consisting of humoral and cellular components. The humoral system comprises lipid precursors known as eicosanoids, the prophenoloxidase system, antimicrobial peptides (AMPs), the hemolymph coagulation system, reactive oxygen species (ROS), and reactive nitrogen species (RNS). The cellular immune system comprises hemocytes, whose function is to phagocytose microorganisms such as bacteria, fungi, and protozoa. Hemocytes are also involved in wound repair by nodulation, in addition to the production of AMPs, RNS, ROS, and prophenoloxidase [4]. Hemocytes additionally have the capacity to express high levels of nitric oxide synthetase, which translates into the production of nitric oxide (NO), a molecule that is part of the constitutive innate immunity in insects [5].

Detailed studies on the saliva of hematophagous arthropods have been performed, focusing on the function of salivary proteins and their role as bioactive molecules that facilitate successful blood feeding, counteracting the coagulation cascade and the complement system of vertebrate immune defense. Hematophagous arthropods have a wide arsenal of proteins with redundant functions involving vasodilatory, antihemostatic, anti-inflammatory, and immunomodulatory activities [6,7].

In the salivary glands, there are also proteins that can stop the infection of pathogens transmitted by these insects, such as *T. cruzi*. Although *T. cruzi* does not directly interact with the triatomine salivary glands because it restricts its life cycle to the insect's intestine, the saliva components that reach the stomach at feeding time may act to kill some genotypes of the parasites [8].

Several studies on parasite–vector interaction have shown the presence of trypanolytic factors (TFs) against some *T. cruzi* DTUs in the hemolymph, midgut, and saliva [8,9,10,11]. TFs have been observed to be differentially expressed among triatomines, which affects the transmission of some *T. cruzi* strains or DTUs, supporting the hypothesis that triatomines are biological filters and modulators of trypanosome transmission [1,10].

Our understanding of the nature of the TFs that are present in the hemolymph and saliva of some triatomine species is still limited. Therefore, the first objective of this study was to confirm the differential lytic activity in the hemolymph and components of salivary glands of *R. prolixus* and *R. colombiensis* against epimastigotes and trypomastigotes of *T. cruzi* I and *T. cruzi* II. The second objective was to carry out a proteomic analysis of the hemolymph and components of salivary glands of these two *Rhodnius* species to identify the immune response proteins possibly related to the observed lytic activity.

## Materials and methods

### Ethical aspects

The Bioethics Committee of the Scientific Research and Development Office at the University of Tolima has granted bioethical approval for the project, aligning with Resolution 008430 of 1993. This resolution establishes scientific, technical, and administrative standards for research and health.

### *Trypanosoma cruzi* strains

To evaluate the lytic activity of the hemolymph and saliva of *R. prolixus* and *R. colombiensis*, reference strains of *T. cruzi* were used: the strain Dm28c representative of DTU I or TcI and the strain Y representative of DTU II or TcII. The parasites were maintained in LIT/NNN biphasic culture medium (Liver Infusion Tryptosa 10% SFB/Novy-McNealk Nicoll) with weekly subcultures.

### Insect colonies

Fifth-instar nymphs of *R. prolixus* and *R. colombiensis* were used. The insects were kept in plastic containers and fed once a week on immobilized live hens. This process takes around 15–20 minutes.These were maintained under a photoperiod of 12 h light/12 h dark at an approximate temperature of 28˚C and relative humidity of 80%.

### Trypanolytic activity of the hemolymph of *R. prolixus* and *R. colombiensis* on cultured epimastigotes of Dm28c (TcI) and Y (TcII)

Following the methodology described by Suarez et al. [10], the insects were fed on chicken blood 8 days before the trypanolytic activity assays were carried out. The hemolymph of 20

insects of each species was collected, mixed, and centrifuged at 14,000 rpm for 5 min. The cell-free supernatant was used to detect trypanolytic activity following the methodology described by Pulido et al. [12]. To prevent melanization of hemolymph, 2 µl of 50 mM phenylthiourea was added to 100 µl of hemolymph. Cultured *T. cruzi* epimastigotes were washed three times with saline solution, centrifuged at 7000 rpm for 5 min, and resuspended in 10% (v/v) LIT medium. A total of 10 µl of hemolymph was added to 10 µl of parasite suspension at a final concentration of $2.5$–$3.5 \times 10^7$ parasites/mL. To confirm the lytic activity, live parasites were counted in a Neubauer chamber at 0 and 14 h of incubation. As a negative control, inactivated hemolymph with 10 µl of pepsin solution (15 mg/mL in 1 M HCl) for every 100 µl of hemolymph was used with subsequent incubation at 37˚C for 4 h. As a positive control, epimastigotes of strain Y were used, which always presented lysis after incubation with the hemolymph of *R. prolixus*. Similarly to sample treatment, both negative and positive controls were treated with phenylthiourea. The experiment was performed in triplicate.

### Trypanolytic activity of hemolymph of *R. prolixus* and *R. colombiensis* on metacyclic trypomastigotes of Dm28c (TcI) and Y (TcII)

$1 \times 10^8$ Dm28c and Y strain epimastigote/mL were cultured at 28˚C in 4 mL liver infusion tryptose (LIT) medium, supplemented with 5% heat-inactivated FBS. The parasites were kept in such conditions for 10 to 12 days; a significant increase in metacyclic trypomastigotes was observed during such time due to nutritional stress. The medium was then passed through a negatively-charged, Sepharose-DEAE (diethylaminoethyl resin) column to retain the epimastgotes and separate them from the metacyclic trypomastigotes [13]. To obtain cell-free hemolymph, the methodology described by Suárez et al. [10] and Pulido et al. [12] was used. Evaluation of the resistance or sensitivity of the metacyclic forms of Dm28c and Y was carried out by incubating 10 µL of trypomastigote suspension at a concentration of $2.5$–$3.5 \times 10^7$ per mL and 10 µL of cell-free hemolymph extract. The resistance or sensitivity of the metacyclic forms was evaluated by estimating the number of parasites by counting in the Neubauer chamber at 0 and 14 h of incubation. Each experiment was done in triplicate.

### Trypanolytic activity of components of salivary glands of *R. prolixus* and *R. colombiensis* on cultured epimastigotes of Dm28c (TcI) and Y (TcII)

To evaluate the lytic activity of components of the salivary glands of *R. prolixus* and *R. colombiensis* against strains Dm28c and Y, salivary glands were obtained 8 days post-feeding by manual extraction from *R. prolixus* and *R. colombiensis*. Once the glands had been extracted, they were washed in 0.9% saline solution to avoid contamination with hemolymph. Subsequently, they were perforated to release the saliva, centrifuged at 14,000 rpm for 5 min at 4˚C, and then the supernatant containing the saliva was recovered.

Incubations were performed with 10 µL of fresh saliva and 10 µL of culture forms of *T. cruzi* suspended in LIT (10% FBS), leaving a final concentration of $2.5$–$3.5 \times 10^7$ parasites/mL. To assess the sensitivity of strains Dm28c and Y to lysis, live parasites were counted in the Neubauer chamber at 0 and 10 h post-incubation. Four replicates were performed for each experiment. As a negative control, LIT (10% FBS) was used instead of fresh saliva. Although conducting experiments using trypomastigote forms and insect saliva was considered, limitations had to be faced regarding the available biological material. Collecting saliva from a single insect only produces 1–2 µL and the insect must also be sacrificed; the experiment would thus have needed a minimum of 60 insects per species.

## Statistical analysis of trypanolytic activity in hemolymph and saliva

Parasite counts were presented as means ± standard deviation and were graphing using the Graphpad Prism 8.0 Program software. The design was aimed at comparing the treatments and controls of incubations. The measurements are repeated, since the same individuals were measured at two different times. For this analysis, a comparison of paired means was made, taking as pairs the measurements of the two times for each treatment. Second, the differences calculated in the previous step were analyzed by one-way ANOVA in which the 5 treatments are compared to each other, using the Tukey test. All comparisons were made to 5% significance.

## Hemolymph and salivary gland protein sequencing by LC/MS/MS

Hemolymph extraction was performed 8 days after feeding the insects with chicken blood. After a cut had been made in the tarsus of the third leg of the insect, the hemolymph was collected with a micropipette in a 1.5 mL tube, kept on ice, centrifuged at 14,000 rpm for 5 min to collect the cell-free supernatant, and then stored at −70˚C until use.

20 pairs of salivary glands were extracted 8 days after feeding the insects. They were washed three times in saline solution (0.9% NaCl), collected in a microtube, and resuspended in saline solution at a volume of 2 μL per pair of glands.

In order to extract the proteins from hemolymph and salivary glands, the tissues were resuspended in lysis buffer (40 mM Tris-Base, 7 M urea, 2 M thiourea, 4% CHAPS, 1 mM PMSF). Subsequently, the samples were incubated on an ice bed for 30 min, with vortexing for 1 min every 10 min. Finally, the cell lysis products were centrifuged at 14,000 rpm for 30 min at 4˚C and the supernatant was removed and stored at −80˚C until use.

The proteins present in the samples were quantified by the Bradford method, using a calibration curve with serial dilutions of bovine serum albumin. Subsequently, polyacrylamide gel electrophoresis was run under denaturing conditions (SDS-PAGE) at 90 V for 10 min in order to use the gel as a storage matrix. These samples were sent to the Proteomics Platform of the CHU Research Center of the University of Laval in Quebec, Canada, where protein digestion and mass spectrometry analysis coupled to high-performance liquid chromatography (LC-MS/MS) were performed.

## Protein digestion

The proteins were extracted from the gels and plated onto 96-well plates by washing with ultrapure water; protein digestion was carried out according to Shevchenko et al. [14]. with modifications suggested by Havlis et al. [15]. The proteins were reduced in 10 mM DTT and alkylated with 55 mM iodoacetamide (IAA) and digested with 126 nM sequencing grade modified porcine trypsin (Promega Madison, WI) at 37˚C overnight. The digestion products were extracted with 1% formic acid in 2% acetonitrile, followed by 1% formic acid and 50% acetonitrile. The extracts were vacuum centrifuged, dried and then suspended in 12 μl 0.1% formic acid. Liquid chromatography with tandem mass spectrometry (LC-MS/MS) was used for analysing 5 μl of the extracts.

## LC-MS/MS analysis

Reversed-phase capillary nanoliquid chromatography (nanoLC) was used for separating the peptides in the samples, which were then analysed by electrospray mass spectrometry (ES-MS/MS). An Agilent 1200 nanopump connected to a 5600 mass spectrometer (AB SCIEX, Framingham, MA, USA), equipped with a nanoelectropulverisation ion source, was used for the

experiments. The peptides were separated on a half-packed PicoFrit column (New Objective, Woburn, MA) (3U, 100 A C18, 15 cm x 0.075 mm internal diameter); they were then eluted with a 5–35% linear gradient with dissolvent B (acetonitrile, 0.1% formic acid) for 64 min at 300 nl/min. The mass spectra were obtained using the data-dependent acquisition mode with Analyst software (SCIEX, version 1.7). Each full scan of the mass spectra (400 at 1,250 m/z) was followed by the dissociation induced by the collision of the 20 most intense ions. Dynamic exclusion was carried out for 12 s at 100 ppm tolerance.

### Identification of proteins from hemolymph and salivary glands of *R. prolixus* and *R. colombiensis*

For the identification of proteins from the hemolymph and salivary glands, the UniProt Triatominae database was used. The MGF files with the list of peaks were obtained with the software (ABSciex), using the Paragon and Progroup algorithms [16]. Subsequently, these files were analyzed using Mascot (Matrix Science, London, UK; version 2.5.1). A value of 0.1 Da was set for the peptide mass tolerance and for the fragment mass tolerance. As fixed modifications, carbamidomethylation of cysteines was established, while as variable modifications, deamination of asparagine and glutamine and oxidation of methionine were included. The information obtained from the identified proteins was visualized through Scaffold version 4.8.3 software, validating peptides and proteins with a false discovery rate (FDR) of less than 1%.

### Quantitative analysis of *R. prolixus* and *R. colombiensis* hemolymph and salivary proteins involved in the immune response

The hemolymph and salivary proteins involved in the immune response were filtered and a semiquantitative profile of the relative abundance of the proteins in both species was created using the label-free method. The normalized spectral abundance factor (NSAF) was used to analyze the spectral count of the three replicates. The calculation obtained with Scaffold is represented by the following expression:

$$SAF = \frac{Exclusive\ spectrum\ number}{protein\ length\ (aa)}$$

The SAF value is normalized using Scaffold's regular quantitative value normalization scheme which takes into account the sum of the SAF values of the analyzed proteins:

$$NSAF = \frac{SAF}{\sum SAF}$$

## Results

### Effects of hemolymph from *R. prolixus* and *R. colombiensis* on epimastigotes and metacyclic trypomastigotes of strains Dm28c (TcI) and Y (TcII)

The incubation of the hemolymph of *R. prolixus* with epimastigotes and metacyclic trypomastigotes of the strain Y showed significant decreases of the parasites at 14h post-incubation as a consequence of parasite lysis (Fig 1A and 1C). The Incubation of *R. prolixus* hemolymph, against epimastigotes and trypomastigotes of the Dm28c strain did not show a significant decrease or lytic activity of the parasites.

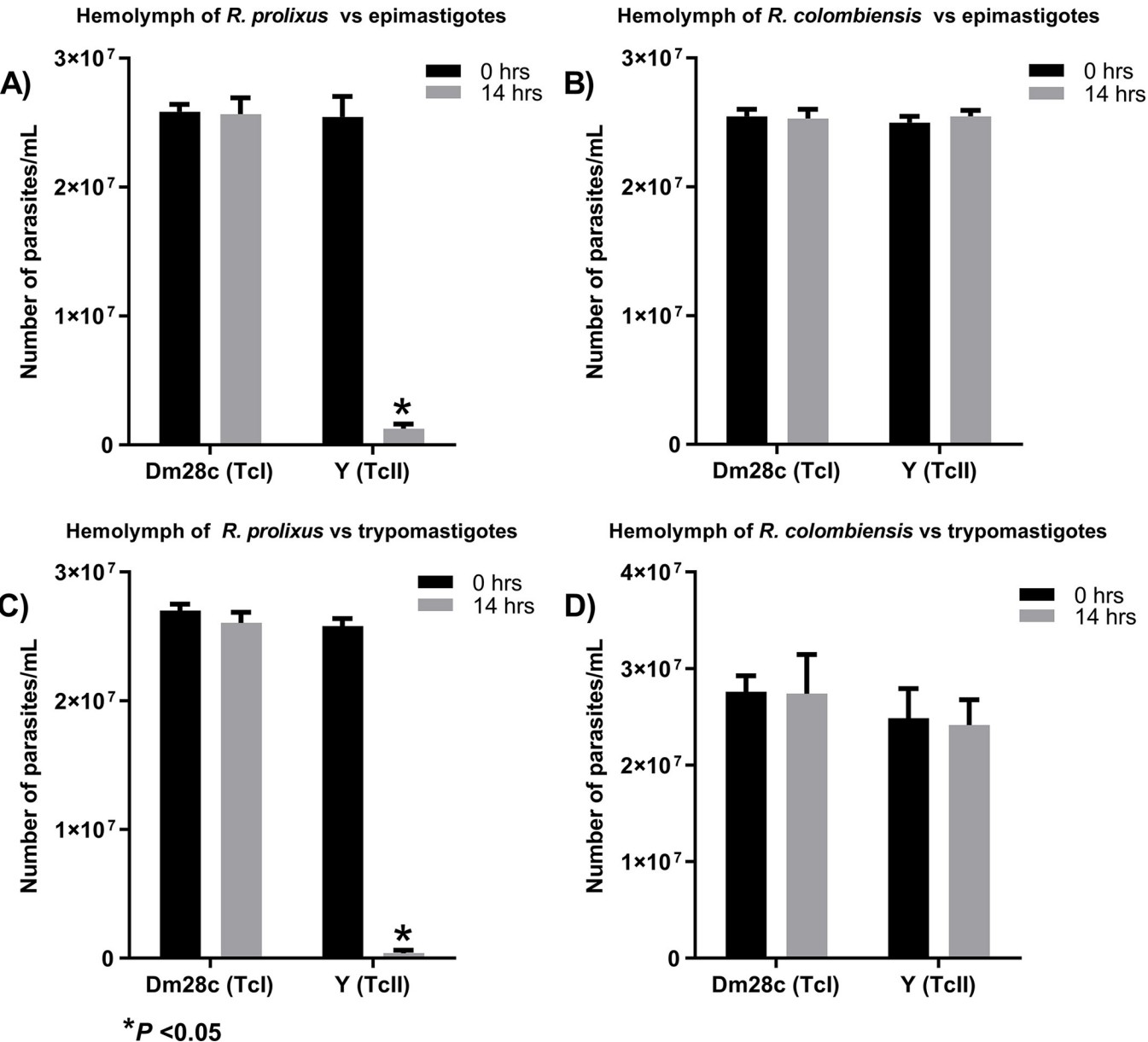

**Fig 1. Incubation of the hemolymph of *R. prolixus* and *R. colombiensis* with epimastigotes and metacyclic trypomastigotes of Dm28c (TcI) and Y (TcII).** The negative control for each experiment showed no lysis or a significant decrease in parasite numbers. The positive control of each experiment showed lysed parasites with a significant decrease in parasite numbers (S1 Table).

The incubation of *R. colombiensis* hemolymph did not show a significant decrease in the number of metacyclic epimastigotes or trypomastigotes of Dm28c or Y during 14 h of incubation; therefore, this study concluded that there was no lytic activity during this time in this vector (Fig 1B and 1D).

## Effects of *R. prolixus* and *R. colombiensis* salivary glands components on epimastigotes of the strains Dm28c (TcI) and Y (TcII)

The results showed lytic activity of the components of salivary glands of *R. prolixus* against Y, with the abundance of parasites showing a significant decrease at 10 h post-incubation, nor

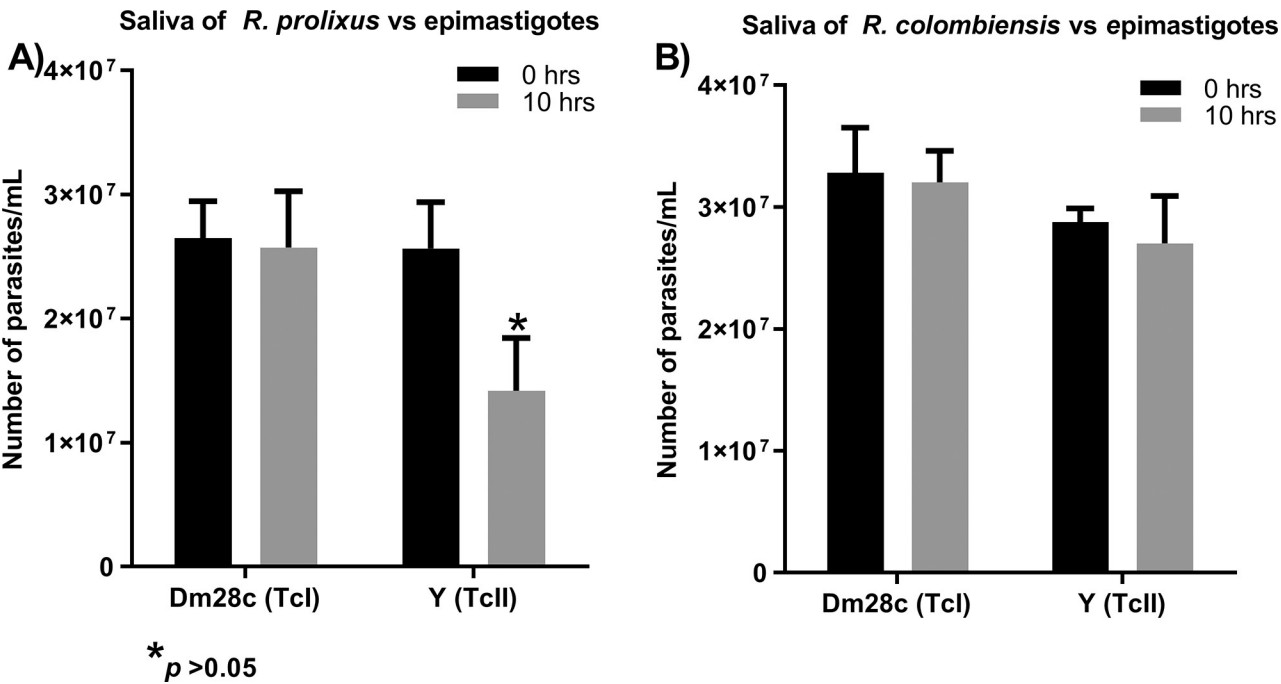

**Fig 2. Incubation of *R. prolixus* and *R. colombiensis* components of salivary glands with epimastigotes of Dm28c (TcI) and Y (TcII).** The negative control for each experiment showed no lysis or a significant decrease in parasite numbers. The positive control of each experiment showed lysed parasites with a significant decrease in parasite numbers (S1 Table).

was there any lytic effect against the Dm28c (Fig 2A). The incubations with the saliva of *R. colombiensis* did not show any lytic activity, nor a significant decrease in the number of parasites of the Dm28c and Y during the first 10 h of incubation (Fig 2B).

### Proteomic analysis of the hemolymph of *R. prolixus* and *R. colombiensis*

A total of 120 proteins were identified in *R. prolixus* hemolymph and 107 in *R. colombiensis* hemolymph (S2 Table). These two species shared a total of 92 proteins. Additionally, 28 proteins were detected only in *R. prolixus* hemolymph and 15 only in *R. colombiensis* (S2 Table).

Of the total proteins identified in the hemolymph of *R. prolixus* and *R. colombiensis*, 40 were associated with an immune response and were grouped into six functional categories that are presented in Fig 3. Quantitative profiling was performed on these proteins involved in the immune response, with their relative abundance based on NSAF. Most of the proteins shared by *R. prolixus* and *R. colombiensis* are involved in carbohydrate and lipid recognition, activation of proteolytic cascades, indicating the presence of common pathogen recognition mechanisms and its products, and mechanisms of melanization and encapsulation through the activation and regulation of the prophenoloxidase system. The relative abundances of proteins of the prophenoloxidase system (A0A1B2G381, A0A1B2G385, T1HW22) were similar in the two species (Fig 3).

Among the proteins related to the metabolism of NO and superoxide ions, we found the nitrophorins Q7YT15 and Q94734, and a putative superoxide dismutase protein (A0A0P4VG48) only in the hemolymph of *R. prolixus*. Nitrophorins are expressed mainly in salivary glands; however, they can be found in other tissues such as testicles, ovary, intestine, Malpighian tubules, and fat bodies [17]. They may also reach other tissues because the hemolymph interacts with all of the organs of the insect due to its open circulatory system. PAMs

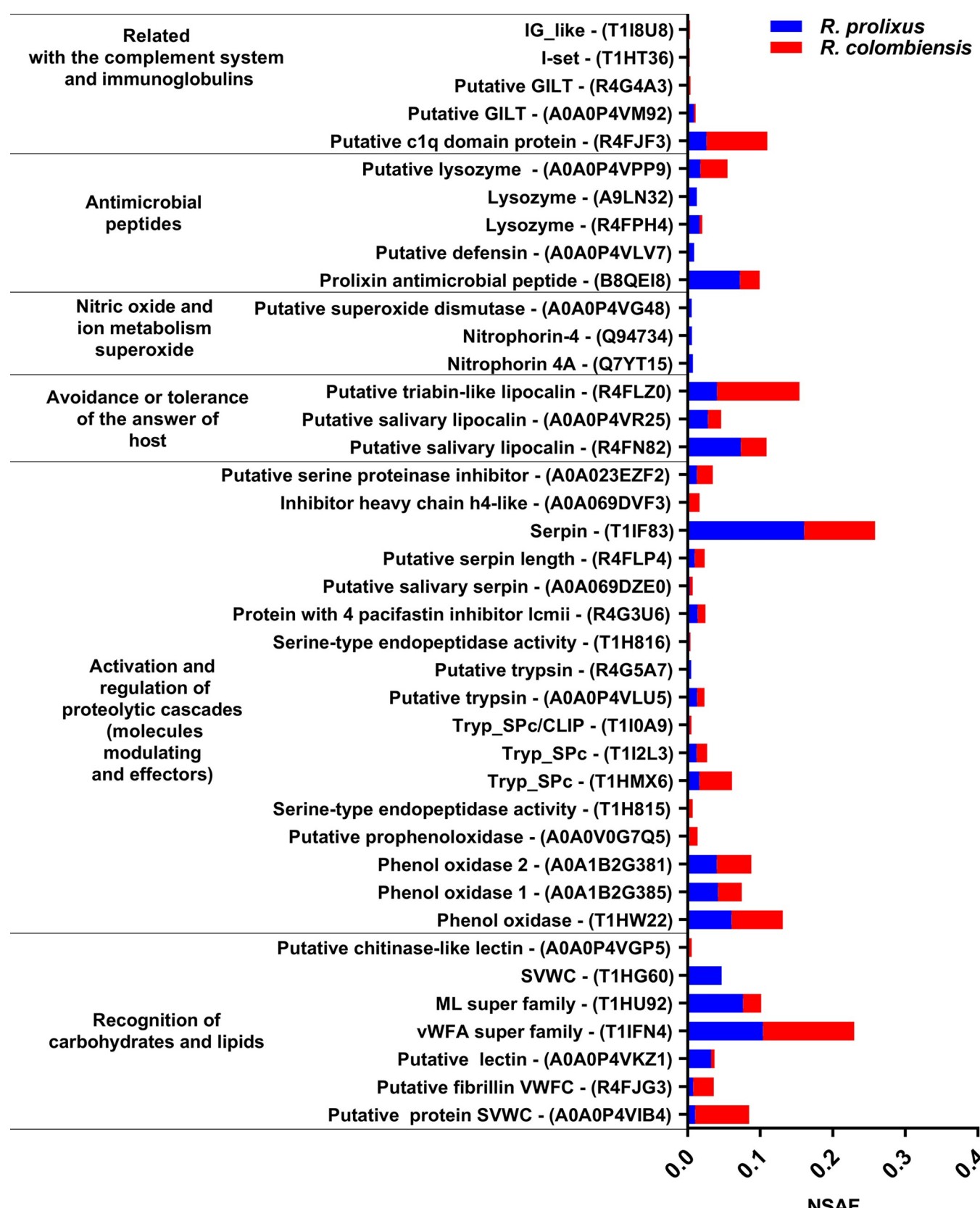

**Fig 3. Relative abundance of 40 proteins involved in immunity in the hemolymph of *R. prolixus* and *R. colombiensis*.** Of these proteins, 32 were shared between the two species, seven were detected only in *R. prolixus*, and one was detected only in *R. colombiensis* (S2 Table).

with higher relative abundance were also detected in *R. prolixus*. These PAMs and proteins such as serine protease with the CLIP domain (T1I0A9), interferon gamma, and superoxide dismutase (A0A0P4VG48) are proteins related to the induced immunity of the insect; that is, they are expressed only after the host has been exposed to infection.

## Proteomic analysis of salivary glands of *R. prolixus* and *R. colombiensis*

A greater number and more diverse functions of proteins were identified in *R. prolixus* than in *R. colombiensis*, with totals of 1103 and 853, respectively (S3 Table). In the salivary glands of both species, 748 proteins were shared, while 355 were detected only in *R. prolixus* and 105 in *R. colombiensis* (S3 Table). Overall, 67 proteins involved in immune activity in the saliva of these insects were classified into four categories to perform a comparative analysis between the two species. The proteins with the highest relative abundance were the nitrophorins (NPs), with the highest representation in *R. prolixus*. In this species, 21 NPs were quantified, compared with 11 in *R. colombiensis*. In the category of evasion or tolerance of the host response, *R. prolixus* presents relative quantification for 38 proteins compared with 10 for *R. colombiensis* (Fig 4). The results of the proteomic characterization in the present work show a more identified NPs and their higher expression of these proteins in *R. prolixus* from the semiquantitative analysis.

## Discussion

### Lytic activity and proteomic analysis of hemolymph

Alvarenga & Bronfen [18] made the first observation of lytic activity against *T. cruzi* in the hemolymph in two triatomine species: *Dipetalogaster maxima* and *Triatoma infestans*. These researchers revealed that the parasites inoculated into the hemocoel of the insects did not survive after a few days, evidencing the inability of *T. cruzi* to establish itself in hemolymph. Meanwhile, Mello et al. [19] showed lytic activity in the hemolymph of *R. prolixus* against strains Dm28c and Y of *T. cruzi*; when these were inoculated in the hemocoel of *R. prolixus*, they were rapidly eliminated. Moreover, via in vitro experiments, Suarez et al. [10] evidenced TFs at the hemolymph of *R. prolixus* and *R. robustus* against DTUs II, V, VI, Tcbat, and *T. cruzi* marinkellei after 14 h of incubation. However, when evaluating the hemolymphs of six more species (*R. colombiensis*, *R. pallescens*, *R. pictipes*, *T. dimidiata*, *T. maculata*, and *P. geniculatus*), none presented in vitro lytic activity against *T. cruzi* DTUs, after 14 h of incubation. In the present work, the hemolymph of *R. prolixus*, in addition to having lytic activity against epimastigotes of Y strain, was confirmed to also lyse the metacyclic trypomastigotes of Y strain, but not those of Dm28c. Regarding the origin of these TFs, they are considered to be part of the remaining innate immunity generated against the intestinal microbiota that would affect some *T. cruzi* genotypes [1,10].

The proteomic results provided our first approach to describing such immune factors. The total amount of proteins identified in this work (120 in *R. prolixus* and 107 in *R. colombiensis*) was less than that reported in other work, such as that by Ouali et al. [20], who mentioned the detection of up to 269 constitutively expressed proteins.

We consider that two variables could explain our lowest value regarding the total number of proteins identified: i) our filter for considering that a protein had been detected in the tissue took the number of *peptide sequences unique to a protein group* as being at least 2, whilst the

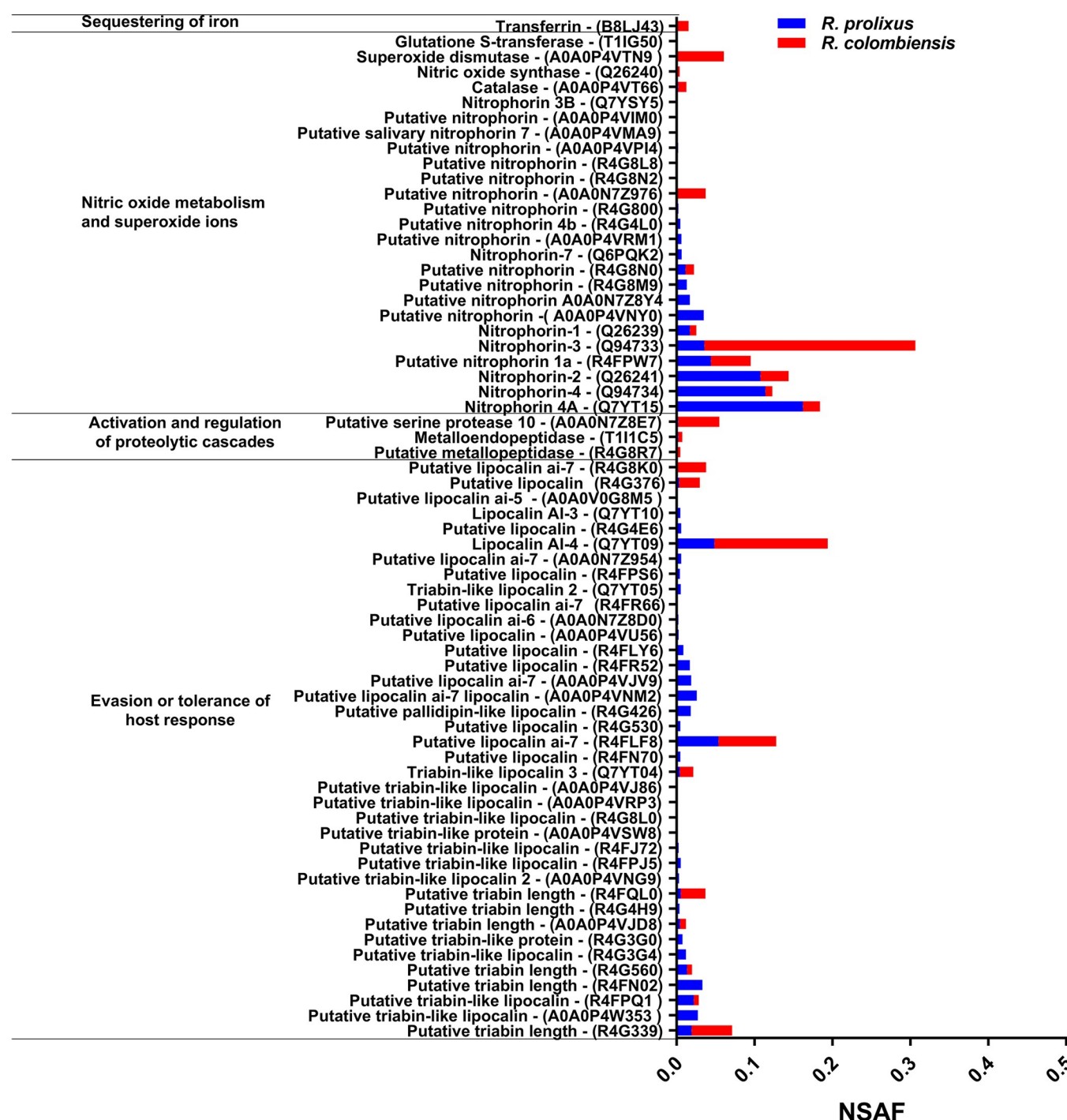

**Fig 4. Relative abundance of 67 proteins involved in the immune response of *R. prolixus* and *R. colombiensis* in the salivary glands.** Of these proteins, 25 were shared between the two species, 41 were detected only in *R. prolixus*, and one was detected only in *R. colombiensis* (S3 Table).

work by Ouali et al. [20] reported proteins having a *unique peptide number* from 1 onwards. *ii*) Our methodology required the haemolymph to be centrifuged to eliminate any cell type. We thus expected that fewer proteins would be detected or at least, only those that are constitutively secreted. 40 proteins related to immunity were the object of interest in our study; this

number of proteins was close to that reported by Ouali et al. [20] who mentioned 58, such figure considered that they also included some proteins whose expression is induced.

Fig 3 shows the relative abundance of immunity-related proteins detected in the hemolymph of *R. prolixus* and *R. colombiensis*. In accordance with the immune factors previously described in *R. prolixus*, the lytic factors observed against Y strain could be associated with AMPs, proteins involved in the metabolism of the prophenoloxidase system, proteins related to hemolymph coagulation, ROS-generating proteins, and protein RNS generators.

Of the proteins related to the activation and regulation of proteolytic cascades (Fig 3), the prophenoloxidase system stands out for the production of melanin, which functions in tissue repair and the encapsulation of pathogens [2,4]. Throughout melanin production, a cascade of free radicals such as ROS and RNS are generated, which are highly toxic against pathogens such as trypanosomes [21]. However, in the lytic activity experiments with hemolymph in this study, phenylthiourea was used as an inhibitor of the prophenoloxidase cascade, and thus the lysis observed in our experiments was not related to the prophenoloxidase system. This supported the assertion that other proteins different from those involved in the prophenoloxidase system additionally act as factors with trypanolytic activity.

Another protein in hemolymph related to the activation and regulation of proteolytic cascades that stands out for its relative abundance is serpin (T1F83). Serpin is responsible for regulating protease activity and therefore oxidative activity because it acts as an inhibitor of proteases, which activate the pathways of the prophenoloxidase system. This function would have a protective role for the insect against an excess of cellular oxidative activity [4,21].

Within the category of carbohydrate recognition, some lectins and the A and C domains of von Willebrand factor were identified in both *R. prolixus* and *R. colombiensis*. Several lectins are conserved in Hemiptera and participate in the defense against flagellates in triatomines [22,23], it has been shown that these binding molecules can induce the recruitment of hemocytes for the encapsulation and melanization of pathogens [24]. Otherwise, as some authors have pointed out, the agglutination processes mediated by these proteins could have a protective effect on parasites, promoting their survival and multiplication [23,25,26]. Because the agglutinating and protective effects of lectins depend on the affinity for the glycoproteins present in the parasite membrane, the affinity for sugars of the detected lectins needs to be examined to confirm their possible protective effect on the different *T. cruzi* DTUs.

The role of AMPs such as lysozyme, defensin, and prolixin cannot be ruled out in trypanolytic activity of hemolymph, because they were more abundant in *R. prolixus* than in *R. colombiensis*. AMPs can alter the structure of the cytoplasmic membrane, generating ion channels that increase its permeability and subsequently induce cell death [27]. The composition of amino acids, their net charge (generally cationic), and their amphipathic and size characteristics promote their interaction with lipid bilayers, mainly those that form the cytoplasmic membranes of pathogens (bacteria, fungi, enveloped viruses, and parasites). Although few studies have focused on the effect of antimicrobial peptides on parasites, some have shown that these molecules can affect their development and trigger cell lysis. Magainin 2 was one of the first AMPs described to show antiparasitic activity, specifically against protozoa. Tests carried out with this peptide in *Paramecium caudatum* led to the lysis of this microorganism [28]. In *Phlebotomus duboscqi*, a defensin active against promastigotes of *Leishmania major* was identified [29]. Additionally, a recombinant attacin from *Glossina* was shown to have trypanolytic activity on *T. brucei* blood trypomastigotes and epimastigotes in vitro and in vivo [30]. The negative effect of antimicrobial peptides on *T. cruzi* has also been demonstrated, since Fieck et al. [31] observed trypanocidal activity of four antimicrobial peptides (apidaecin, magainin II, melittin, and cecropin) on *T. cruzi*, even at concentrations where they had no effect on *Rhodococcus rhodnii*. Subsequently, the combined treatment of these peptides increased the toxicity on the parasites.

An interesting finding in the hemolymph of *R. prolixus* was the detection of NPs and lipocalins which are known to be synthesized in the salivary glands of insects; however, they may reach the hemolymph because it interacts with all of the insect's organs due to its open circulatory system. These proteins are related to the function of facilitating insect feeding when it takes blood from its host because they have vasodilatory and anticoagulatory properties. Any type of salivary gland-related contamination during haemolymph extraction was ruled out since tissue collection was carried out independently. Firstly, the haemolymph was collected; the insect was kept alive during such procedure and only a cut in one of its hind legs was needed, as mentioned in the methodology, so that the salivary glands suffered no risk of rupture during haemolymph collection at any time.

Other research has already identified NPs in *R. prolixus* haemolymph. Ouali et al. [20] analysed haemolymph protein expression regarding *T. cruzi* infection and, although these proteins' role concerning insect immunity was not discussed regarding the objective of their work, it can be seen that 4 NP identifications were reported (Q7YT15, Q26239, T1HKP3, R4G8M6). A recent study by Santos et al. [17] on lipocalins (which included nitrophorins) mentioned that such proteins could be present in other types of tissue and could have multiple functions. The authors suggested that central nervous system expressed lipocalins the can act as neurotransmitters, as odorant-binding proteins (OBPs) when expressed in antennae and as lipid transporters when expressed in the haemolymph.

*Rhodnius prolixus*'s specific lytic activity against the *T. cruzi* Y strain, but not against the *T. cruzi* dm28C strain, is probably the result of the overexpression of genes such as nitrophorins (NPs). These proteins are involved in the metabolism of NO, a free radical that acts in the constitutive innate immunity of the insect [32].

Nitrophorin 4-A (NP4) was found in *R. prolixus* haemolymph in this work; NP4 is important because nitric oxide (NO) from nitrophorins could be released in the foregut and act on the genotypes of parasites which are sensitive to oxidative stress. *In vivo* experiments have shown that increased NO production in *R. prolixus* hemolymph and digestive tract has correlated with decreased in *T. cruzi* multiplication; on the contrary, the parasite manages to maintain and multiply itself when NO production becomes decreased [33].

Although the mechanism regarding how NPs are expressed in haemolymph remain unknown, it is likely that they are secreted directly into the haemocoel as a factor of humoral immunity [34]. Other immune factors, such as antimicrobial peptides, reactive nitrogen and oxygen intermediates and complex enzymatic cascades (such as the prophenoloxidase system) contributing to coagulation or melanisation in the haemolymph, also act as immune response components for eliminating potential pathogens acquired during feeding [34].

## Lytic activity and proteomic analysis of salivary glands

Fig 4 shows the relative abundance of proteins in the salivary glands of *R. prolixus* and *R. colombiensis*. A large number of triabins, lipocalins, and nitrophorins are more abundant in *R. prolixus* than in *R. colombiensis*. The main role of salivary proteins in blood-feeding arthropods is to maintain blood flow in the mouthparts that successfully conduct blood to the digestive tract. This process is successful due to the combination of numerous salivary proteins, in some cases small molecules, that act together to inhibit the coagulatory cascade, limit platelet activation, and prevent vasoconstrictive responses. In triatomine salivary glands, there are still many families of proteins that have not been completely characterized and of which several additional activities could be found. According to Arca & Ribeiro [7], up to 40% of salivary peptides in hematophagous insects have unknown functions. When considering only the 155 described species of triatomines, there is proteomic information for just 16 species, supported

by nine annotated sialotranscriptomes, six descriptive sialoproteomes, and seven sialomes [35–46]. Added to this, in each of these studies, a large number of proteins were obtained without being able to characterize them. Within the reports on these studies, transcriptomic and proteomic data for *R. prolixus* are presented [36]. For *R. colombiensis*, this report presents the first proteomic data on hemolymph and salivary glands.

The above-mentioned studies focused almost exclusively on the analysis of salivary proteins related to anticoagulant, antiplatelet, and vasodilatory activities to respond to the hemostasis of their vertebrate host, properties that could have pharmacological potential. The role that these salivary proteins may have in the immunity of triatomines has not been discussed in depth, despite there being evidence of them having antiparasitic, antibacterial, antiviral, and antifungal activities [7,46].

The results of the present work on the effect of *R. prolixus* salivary proteins on *T. cruzi* epimastigotes and trypomastigotes showed lytic activity against Y strain. This effect is similar to that observed in an experiment carried out by Ferreira et al. [8] using the content of the salivary glands of *R. prolixus*, which showed lysis of 20% of the trypomastigote forms of *T. cruzi* (strain CL). The results of these experiments indicate that the proteins present in the salivary glands in *R. prolixus*, in addition to fulfilling the functions that counteract the hemostasis of their vertebrate host, can also modulate the infection and adaptation of pathogens and particularly some DTUs of *T. cruzi* [8,9]. It might be thought that the effect of these lytic factors would not be relevant to *T. cruzi* due to their absence from salivary glands during their life cycle in the vector. However, part of the saliva that is ingested in the insect's feeding process is known to reach the intestine and thus interacts directly with the parasite. This innate immune response generated in the salivary glands has been reported to affect some genotypes of *T. cruzi* [1].

The proteins in the salivary glands of triatomines that are related to immune functions against pathogens include antimicrobial peptides, lysozyme, pattern recognition molecules, and serine proteases, which act as activators of the prophenoloxidase system [4,7,47]. A pore-forming lytic protein called trialysin was identified in the saliva of *Triatoma infestans*, which lysed the trypomastigote forms of *T. cruzi* II (strain Y) [9]; however, no protein with similar characteristics in the salivary glands of *R. prolixus* has been identified. Although there is evidence of lytic activity against *T. cruzi* in the salivary glands of *R. prolixus*, the factors involved in this lysis have remained unclear. We know that this lytic effect against Y strain observed in the salivary glands of *R. prolixus* has also been observed in the hemolymph of *R. prolixus* and *R. robustus*, while being absent from the salivary glands and hemolymph of *R. colombiensis* and the hemolymph of *R. pallescens* [10]. In this sense, the question arises about the epidemiological role of this lytic factor, which would only be present in the salivary glands of some *Rhodnius* species.

In *R. prolixus* and *R. colombiensis*, proteins involved in NO metabolism and therefore in ROS metabolism were found. The NO and ROS molecules are considered to be constitutive immune components conserved in Hemiptera and thus they are relevant factors in the defense of triatomines [5]. Although similar proteins were identified in both species in relation to NO metabolism, such as the enzyme nitric oxide synthase (Q26240), *R. prolixus* presented a greater diversity of lipocalins and nitrophorins that generate a greater machinery of oxidative activity that reinforces its innate immune response [48,49].

In different studies on triatomine sialoma, it has been shown to contain a predominance of lipocalins, triabins, and NPs [37,41,50]. In *R. prolixus*, the lipocalin family presents a very significant component compared with the rest of the proteins present in saliva [48]. Specifically, reference has been made to the great abundance of NPs in the saliva of *R. prolixus* [48,51]. NPs have been very well characterized at the structural and biochemical levels. The main function of NPs is related to the transport, storage, and release of NO. These molecules are considered

cytotoxic factors against *T. cruzi*, and pathways involving the radical activity of ROS develop around NO metabolism, also act against parasites [6,32,52,53]. Those lipocalins and nitrophorins with higher relative abundance in *R. prolixus* than in *R. colombiensis* are candidate factors responsible for the lysis observed.

## Differential immune response of *R. prolixus* y *R. colombiensis*

Several studies have indicated that the immune response of triatomines plays an important role in the success or failure of transmission of some *T. cruzi* DTUs. In the hemolymph and saliva of some *Rhodnius* species, there are proteins that activate oxidative mechanisms that can inhibit the infection of some *T. cruzi* DTUs. In this study, a comparative proteomic analysis of the hemolymph and salivary proteins of *R. prolixus* and *R. colombiensis* was performed for the first time. This analysis showed the relative abundance of nitrophorins in *R. prolixus*, which act together with other proteins such as lysozyme, prolixin, lipocalins, and triabins to generate a strong immune response in *R. prolixus*. This response should be responsible for the lytic activity of hemolymph and saliva against epimastigotes and trypomastigotes of *T. cruzi* II, detected in vitro. These findings complement the observations of lytic activity of hemolymph on *T. cruzi* V, *T. cruzi* VI, *T. cruzi* bat, and *T. cruzi* marinkellei reported by Suárez et al. [10].

The resistance of strain Dm28c to the lysis of *R. prolixus* hemolymph and saliva proteins observed in this work, together with the results of Suarez et al. [10], where they analyzed 20 different TcI strains with the same result, allows us to conclude that resistance would be occurring at the DTU I level, due to possible mechanisms that allow it to evade oxidative stress. *T. cruzi* I is the DTU with the widest geographical distribution, from the southern United States to the center of Chile and Argentina, a distribution that could be related to the resistance to oxidative stress of the vectors.

Similarly, we can say that lysis against strain Y could occur at the level of DTU II. The vigorous immune response observed in *R. prolixus* against *T. cruzi* II was also observed in *R. robustus* [10] and could be a determinant of the vectorial inability of these species to transmit *T. cruzi* II. Studies carried out with *R. robustus* showed its inability to transmit *T. cruzi* II in experimental infections [54]. Meanwhile, studies carried out in Colombia did not detect *T. cruzi* II in the *R. prolixus* specimens examined [55,56].

The genus *Rhodnius* is made up of 21 species divided into three groups: the Pallescens group with three species (*R. colombiensis*, *R. ecuadoriensis*, *R. pallescens*) [57], in which in vitro assays have not detected trypanolytic factors in hemolymph or saliva; the Pictipes group with seven species (*R. amazonicus*, *R. brethesi*, *R. micki*, *R. paraensis*, *R. pictipes*, *R. stali*, *R. zeledoni*), in which in vitro tests have not been carried out to verify the presence of trypanolytic factors in hemolymph or saliva; and the Prolixus group, with 11 species, of which *R. prolixus* and *R. robustus* present trypanolytic factors in hemolymph and salivary glands. Therefore, new studies are needed to verify the presence of this vigorous immune response in the remaining nine species of the Prolixus group (*R. barretti*, *R. dalessandroi*, *R. domesticus*, *R. milesi*, *R. marabaensis*, *R. montenegrensis*, *R. nasutus*, *R. neglectus*, *R. neivai*).

Despite the limitations of proteomic studies, related to reproducibility, analysis, and identification of a high number of proteins, the present work was able to show differences in the relative abundance of proteins involved in the immune response of *R. prolixus* and *R. colombiensis*, which could be associated with the lytic activity observed in the hemolymph and salivary glands of *R. prolixus* against epimastigotes and trypomastigotes of strain Y (TcII), but not against Dm28c (TcI).

To more precisely identify the proteins involved in this immune response, new comparative transcriptomic studies in triatomine species with and without lytic activity in hemolymph and

salivary glands should be carried out, and the expression of proteins possibly involved in this immune response by quantitative PCR needs to be evaluated. Meanwhile, studying the interaction of the intestinal microbiota of the vectors with the parasites and investigating the mechanisms of resistance to oxidative stress in the DTUs of *T. cruzi* (*T. cruzi* I–VI and *T. cruzi* bat) are necessary, to understand innate immunity, parasite–vector interaction, and coevolution of parasites and their vectors. Further study and investigation should then clarify the uneven geographical distribution of DTUs associated with the complex epidemiology of Chagas disease in different parts of the Western Hemisphere.

## Supporting information

**S1 Table. Counts of parasite incubations with the saliva and hemolymph of *R. prolixus* and *R. colombiensis*.**
(DOCX)

**S2 Table. Common proteins detected of *R. prolixus* and *R. colombiensis* hemolymph, detected only in *R. prolixus* hemolymph, or detected only in *R. colombiensis*.**
(DOCX)

**S3 Table. Common proteins detected of *R. prolixus* and *R. colombiensis* salivary glands, detected only in *R. prolixus salivary glandas*, or detected only in *R. colombiensis*.**
(DOCX)

## Acknowledgments

We thank Edanz Group (https://www.edanz.com/ac) for editing a draft of this manuscript.

## Author Contributions

**Conceptualization:** Hamilton J. Barbosa, Yazmin Suárez Quevedo, Arlid Meneses Torres, Gustavo A. Vallejo.

**Data curation:** Hamilton J. Barbosa, Yazmin Suárez Quevedo, Arlid Meneses Torres, Gustavo A. Vallejo.

**Formal analysis:** Hamilton J. Barbosa, Yazmin Suárez Quevedo, Arlid Meneses Torres, Gustavo A. Vallejo.

**Funding acquisition:** Hamilton J. Barbosa, Yazmin Suárez Quevedo, Gustavo A. Vallejo.

**Investigation:** Hamilton J. Barbosa, Yazmin Suárez Quevedo, Arlid Meneses Torres, Gustavo A. Gaitán Veloza, Julio C. Carranza Martínez, Daniel A. Urrea-Montes, Carlos Robello-Porto, Gustavo A. Vallejo.

**Methodology:** Hamilton J. Barbosa, Yazmin Suárez Quevedo, Arlid Meneses Torres.

**Project administration:** Hamilton J. Barbosa, Julio C. Carranza Martínez, Daniel A. Urrea-Montes, Gustavo A. Vallejo.

**Resources:** Hamilton J. Barbosa, Arlid Meneses Torres, Gustavo A. Gaitán Veloza, Gustavo A. Vallejo.

**Supervision:** Julio C. Carranza Martínez, Daniel A. Urrea-Montes, Gustavo A. Vallejo.

**Writing – original draft:** Hamilton J. Barbosa, Yazmin Suárez Quevedo, Gustavo A. Vallejo.

**Writing – review & editing:** Hamilton J. Barbosa, Yazmin Suárez Quevedo, Arlid Meneses Torres, Gustavo A. Gaitán Veloza, Julio C. Carranza Martínez, Daniel A. Urrea-Montes, Carlos Robello-Porto, Gustavo A. Vallejo.

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
