## [Decision Letter · Decision Letter 0]

2 Oct 2023

Dear Sr. Barbosa,

Thank you very much for submitting your manuscript "Comparative proteomic analysis of the hemolymph and salivary glands of Rhodnius prolixus and R. colombiensis reveals candidates associated with differential lytic activity against Trypanosoma cruzi I and T. cruzi II" for consideration at PLOS Neglected Tropical Diseases. As with all papers reviewed by the journal, your manuscript was reviewed by members of the editorial board and by several independent reviewers. In light of the reviews (below this email), we would like to invite the resubmission of a significantly-revised version that takes into account the reviewers' comments. 

After revision of your manuscript, I would invite the authors to fully address the comments from the reviewers.

The MS is original and scientifically sound but require substantial revision prior new round of analysis.

All of the reviewers comments and suggestions must be properly addressed and answered.

We cannot make any decision about publication until we have seen the revised manuscript and your response to the reviewers' comments. Your revised manuscript is also likely to be sent to reviewers for further evaluation.

Sincerely,

Edmundo Carlos Grisard, BSc, PhD

Guest Editor

Ricardo Fujiwara

Section Editor

Dear Dr. Barbosa,

After criterion's revision of you manuscript, I would invite the authors to fully address the comments from the reviewers.

The MS is original and scientifically sound but require substantial revision prior new round of analysis.

All of the reviewers comments and suggestions must be properly addressed and answered.

Reviewer's Responses to Questions

**Key Review Criteria Required for Acceptance?**

**Methods**

-Are the objectives of the study clearly articulated with a clear testable hypothesis stated?

-Is the study design appropriate to address the stated objectives?

-Is the population clearly described and appropriate for the hypothesis being tested?

-Is the sample size sufficient to ensure adequate power to address the hypothesis being tested?

-Were correct statistical analysis used to support conclusions?

-Are there concerns about ethical or regulatory requirements being met?

Reviewer #1: (No Response)

Reviewer #2: The primary objective was to confirm the differential lytic activity in the hemolymph and components

 of salivary glands of R. prolixus and R. colombiensis against epimastigotes and trypomastigotes of T. cruzi I and T. cruzi II. This cannot be confirmed using only a single strain per DTU

- Authors failed to mention the repository where the proteomic data was placed.

- Did the authors evaluate the protein integrity in the pepsin treated control hemolymph? (SDS-PAGE gels or otherwise?). It would be of interest if they could specify if they did.

- Did the control haemolymph also contain phenyl thiourea? Please clarify

- Line 182: “10 μL of T. cruzi culture forms containing a concentration of 2.5–3.5 ×107 parasites/mL” what were these parasites suspended in before mixing 1:1 with LIT?

- In line 185: what is 10% LIT? What was the LIT medium diluted in? Or are the authors refering to LIT medium containing 10% FBS? Please clarify

**Results**

-Does the analysis presented match the analysis plan?

-Are the results clearly and completely presented?

-Are the figures (Tables, Images) of sufficient quality for clarity?

Reviewer #1: (No Response)

Reviewer #2: (No Response)

**Conclusions**

-Are the conclusions supported by the data presented?

-Are the limitations of analysis clearly described?

-Do the authors discuss how these data can be helpful to advance our understanding of the topic under study?

-Is public health relevance addressed?

Reviewer #1: (No Response)

Reviewer #2: - T. cruzi is highly variable. Great variability exists within DTUs, specially TcI. Having tested only one strain for each DTU, the authors cannot generalize findings to entire DTUs. The entire manuscript needs to be edited, even figures and figure legends, where labels should read “Dm28c” instead of TcI and “Y” instead of TcII. Currently, the manuscript overgeneralizes conclusions to entire DTUs, which is not supported by the data from a single strain per DTU. Therefore, the conclusions and discussion need to be carefully revised to ensure no overgeneralizations, which extend beyond what is supported by the data, are made.

- T. cruzi infects the vector´s digestive tract. Therefore, during infection of triatomines, the parasite is not in contact with haemolymph. As a result, the fact that the haemolymph displays lytic effect against some strains and not others would not affect the completion of the parasite´s life cycle. Therefore, authors need to explain why is it relevant to study the lytic effect of haemolymph as opposed to, for example, proteins expressed in the vector digestive tract.

- Although the proteomic analysis is interesting, there are no specific experiments connecting the lytic effect in the hemolymph or saliva with specific proteins. The authors only speculate regarding possible factors involved in the lysis, based on the relative abundance of proteins for each vector species.

**Editorial and Data Presentation Modifications?**

Reviewer #1: (No Response)

Reviewer #2: (No Response)

**Summary and General Comments**

Reviewer #1: The study by Barbosa and colleagues entitled “Comparative proteomic analysis of the hemolymph and salivary glands of Rhodnius prolixus and R. colombiensis reveals candidates associated with differential lytic activity against Trypanosoma cruzi I and T. cruzi II” evaluated the trypanolytic effect of hemolymph and saliva proteins of R. prolixus and R. colombiensis, using two strains of T. cruzi, Dm28c and Y. The study also described and compared the hemolymph and saliva proteome between the two Rhodnius species. Finally, they tried to associate the immune proteins found in the proteome with the trypanolytic effects observed in the in vitro experiments. The experimental design is sound and the methodology employed is appropriate and well established. However, there are some issues in methodology and results that should be addressed before the manuscript is accepted for publication. 

Line 122. Check that the strain used was not Dm28c.

Line 129. Describe in more detail how the triatomines were fed, including information on the ethics of animal use.

Line 137. When describing the methodology and results, the authors generalized the strains used as TCI and TCII, but only one strain of each DTU was tested. In addition, as Dm28c and Y are reference strains, which have been maintained in culture medium for a long time, they may not be the best representatives of these DTUs. Therefore, authors should replace TCI with Dm28c and TCII with Y throughout the text.

Line 148. How many biological replicates were used?

Line 159. Were these parasites obtained from culture medium? Include details of this procedure.

Line 168. Were these samples technical or biological replicates?

Line 181. Why did the authors not use trypomastigote forms for this experiment? These are the forms that will be in contact with the triatomine saliva that is ingested during the feeding process.

Line 185. Why is the negative control of saliva assays different from that of hemolymph?

Line 190. Anova is not the correct statistical test to compare these data, which are dependent as they come from the same sample measured at different times. Authors should reanalyze them using a test suitable for repeated samples.

Line 197. What was the number of samples used? Describe how samples were prepared for LC/MS/MS and the conditions used in the mass spectrometer.

Line 198. If the procedure was the same as the previous experiment, describe the details the first time you mention it.

Line 202. How many salivary glands were used?

Line 208. Why was hemolymph exposed to a cell lysis procedure if it was previously centrifuged to remove the cells?

Line 256. I suggest replacing the title with: Effects of hemolymph from R. prolixus and R. colombiensis on epimastigotes…

Line 263. Metacyclic epimastigotes? I suggest revising the language in the results section.

Line 269. Include, at least in the description, the results of the negative and positive controls (same for fig. 2).

Line 280. I suggest replacing the title with: Effects of R. prolixus and R. colombiensis salivary gland components on T. cruzi epimastigotes.

Line 303. In the study by Ouali et al (2022), 376 proteins were identified in the hemolymph of R. prolixus. Include this study on the discussion of these results.

Ouali, R., Vieira, L. R., Salmon, D., & Bousbata, S. (2022). Rhodnius prolixus Hemolymph Immuno-Physiology: Deciphering the Systemic Immune Response Triggered by Trypanosoma cruzi Establishment in the Vector Using Quantitative Proteomics. Cells, 11(9), 1449.

Line 376. I wouldn’t generalize the results to DTU genotypes, as only one strain of each DTU was tested.

Line 444. I cannot understand how a protein that is enclosed in the salivary glands would cross the membrane to reach the hemolymph. In the case they are not synthetized in the hemolymph, could be the presence of these proteins a contamination caused by the rupture of the salivary glands during insect dissection?

Line 547. Did the authors inoculate the Dm28c strain in the hemolymph of R. colombiensis to see how long the parasites survive?

Reviewer #2: In this manuscript, Barbosa et al. evaluated the lytic activity of haemolymph and saliva of R. prolixus and R. colombiensis against T. cruzi. The authors based their conclusions in experiments with the DM28c (TcI) and Y (TcII) strains, where the heamolymph of R. prolixus, but not R. colombiensis was capable of lysing both epimastigotes and tripomastigotes of the Y strain, while those of the Dm28c remained unaffected. Additionally, the authors report a lytic effect by R. prolixus saliva against Y strain epimastigotes. Finally, the authors performed a proteomic analyisis of haemolymph and saliva from the two vectors species to identify candidate lytic proteins. Unfortunately, I feel the data presented is not sufficient to support DTU level differences, only differences for two specific strains (Dm29c and Y) can be established. Overextending the conclusions to encompass the entire DTUs TcI and TcII is not appropriate.

PLOS authors have the option to publish the peer review history of their article (what does this mean?). If published, this will include your full peer review and any attached files.

Reviewer #1: No

Reviewer #2: No
---

## [Editor Report · Decision Letter 1]

7 Mar 2024

Dear Sr. Barbosa,

We are pleased to inform you that your manuscript 'Comparative proteomic analysis of the hemolymph and salivary glands of Rhodnius prolixus and R. colombiensis reveals candidates associated with differential lytic activity against Trypanosoma cruzi Dm28c and T. cruzi Y' has been provisionally accepted for publication in PLOS Neglected Tropical Diseases.

Best regards,

Paul Brindley

EIC

Shaden Kamhawi

EIC

---

## [Editor Report · Acceptance letter]

20 Mar 2024

Dear Sr. Barbosa,

We are delighted to inform you that your manuscript, "Comparative proteomic analysis of the hemolymph and salivary glands of Rhodnius prolixus and R. colombiensis reveals candidates associated with differential lytic activity against Trypanosoma cruzi Dm28c and T. cruzi Y," has been formally accepted for publication in PLOS Neglected Tropical Diseases.

Best regards,

Shaden Kamhawi

co-Editor-in-Chief

Paul Brindley

co-Editor-in-Chief
